# Isolation and Characterization of a Novel Thermostable Bacteriophage Targeting Multi-Drug-Resistant *Salmonella* Enteritidis

**DOI:** 10.3390/v17111518

**Published:** 2025-11-19

**Authors:** Salman A. Almashtoub, Gabriel H. Fares, Tasnime A. Abdo Ahmad, Sara Barada, Ahmad Turk, Dayana Shoukair, Ghassan M. Matar, Esber S. Saba

**Affiliations:** 1Department of Biology, Faculty of Sciences-I, Lebanese University, Beirut P.O. Box 6573, Lebanon; salmanalmashtoub3@gmail.com (S.A.A.); gabrielfares0@gmail.com (G.H.F.); 2Department of Experimental Pathology, Immunology, and Microbiology, Faculty of Medicine, American University of Beirut, Beirut 1107 2020, Lebanon; taa53@mail.aub.edu (T.A.A.A.); sb133@aub.edu.lb (S.B.); at108@aub.edu.lb (A.T.); dayanashoukair@gmail.com (D.S.); gmatar@aub.edu.lb (G.M.M.); 3Center of Infectious Diseases Research CIDR, Faculty of Medicine, American University of Beirut, Beirut 1107 2020, Lebanon

**Keywords:** *Salmonella* Enteritidis, bacteriophage, multidrug-resistant bacteria, foodborne pathogens, lytic phage, thermostable, phage therapy, food safety, biocontrol

## Abstract

(1) Background: The emergence of multidrug-resistant (MDR) *Salmonella enterica* poses a major threat to global public health, underscoring the urgent need for alternative therapeutic strategies. Bacteriophages represent a promising alternative due to their high specificity and potent ability to lyse MDR strains. (2) Methods: In this study, we isolated a novel MDR *Salmonella* Enteritidis-targeting bacteriophage from Lebanese sewage and characterized its host range, thermal and pH stability, and infection dynamics. Whole-genome sequencing was performed using Illumina technology to determine its genetic features and taxonomic classification. (3) Results: the bacteriophage was classified within the genus *Jerseyvirus* and the class *Caudoviricetes* with a 43 kb dsDNA genome encoding 66 open reading frames (ORFs). It demonstrated remarkable thermal stability, retaining infectivity after prolonged incubation at 65 °C, and showed a broad host range. The phage formed large, clear plaques, displayed rapid adsorption (>97% within 3 min), a short latent period (20 min), and a burst size of ~32 PFU per cell. Genome analysis revealed no lysogeny, virulence, or resistance genes, confirming its strictly lytic nature and supporting its potential use as a biocontrol agent. (4) Conclusions: These findings identify SA01 as a novel, strictly lytic, and thermally stable bacteriophage with strong potential as a biocontrol agent against multidrug-resistant *Salmonella* Enteritidis. Its broad host range suggests potential activity also against other *Salmonella* enterica serovars, supporting its applicability in food safety and biotechnology.

## 1. Introduction

Salmonellosis, an infection caused by *Salmonella* spp., is one of the most significant foodborne zoonotic diseases [1], accounting for an estimated 95.1 million cases of gastroenteritis each year [2]. In humans, infection is commonly linked to the consumption of raw or undercooked meat, eggs, and other poultry-derived products [3]. Many *Salmonella enterica* strains are associated with poultry; the avian-adapted biovars *Gallinarum* and *Pullorum* are primarily restricted to birds and rarely infect humans [4] while the serovars Enteritidis and Typhimurium are among the most prevalent in poultry and are major causes of foodborne salmonellosis [5]. Ready-to-eat (RTE) foods are of particular concern, since they are consumed without further cooking, which would normally eliminate *Salmonella.* Processed foods can also become contaminated after production, leading to infection [6,7]. In poultry, *Salmonella* transmission occurs through multiple routes, including contaminated feed, contact with infected animals, and vertical transmission from breeding stock to offspring [8]. Therefore, implementing effective control strategies at both the farm and food production levels is crucial to ensure food safety and public health [9].

Overuse and misuse of antibiotics in medicine and agriculture have accelerated the emergence of antimicrobial-resistant (AMR) strains among these pathogens, limiting the effectiveness of first-line antibiotics and complicating treatment [10,11]. The World Health Organization (WHO) has identified AMR as one of the greatest threats to global public health, estimating that, without effective interventions, it could lead to up to 10 million deaths annually by 2050 [6,12]. In agriculture, antibiotics are often used not only for therapy but also for growth promotion and disease prevention, fostering the development of multidrug-resistant (MDR) bacteria. As a result, resistant strains from different bacterial species, including *Salmonella enterica*, *Escherichia coli*, and *Staphylococcus aureus*, have been detected in dairy products, RTEs, and meat, representing a serious threat to public health and highlighting the need for new hygiene practices and effective control strategies [7,10,12,13].

One of the most promising alternatives to antibiotics is bacteriophages—naturally occurring viruses that selectively infect and lyse bacterial pathogens—highlighting their potential role in eradicating and eliminating resistant strains [14]. The Food and Drug Administration (FDA) has approved bacteriophage products as generally recognized as safe (GRAS), permitting their application in livestock and poultry products [15]. Numerous studies have reported successful reductions in bacterial populations using bacteriophage cocktails applied extrinsically to experimentally contaminated feed [16], raw meat, and vegetables [17], as well as therapeutically administered phages after antibiotic treatment failed [14].

In this study, we isolated and characterized a novel MDR—*Salmonella* Enteritidis—targeting bacteriophage exhibiting exceptional thermal stability. These findings provide a foundation for the development of future bacteriophage-based interventions for therapy, food safety, and biotechnology.

## 2. Materials and Methods

### 2.1. Bacterial Strain and Growth Conditions

The nineteen MDR *Salmonella* Enterica strains used in this study were clinical isolates obtained from Lebanon and were provided by the bacteriology lab, Department of Experimental Pathology, Immunology and Microbiology, American University of Beirut, Lebanon; Table A1, Table A2, Table A3 and Table A4. Bacteria were initially cultured on SS (*Salmonella-Shigella*) agar, followed by subculture on LB (Luria–Bertani) agar. For long-term storage, the stains were kept at −20 °C in 50% glycerol [18].

### 2.2. Screening and Isolation of SA01 Bacteriophage

Sewage samples were collected from twenty-three sites across Lebanon. Each sample was centrifuged at 4000× *g* for 15 min at 4 °C to remove debris, then filtered through a 0.2 µm syringe filter [19]. The filtrates were pooled into eight composite samples for preliminary screening against eight *Salmonella* strains. For primary screening, 50 µL of sewage combination was mixed with 50 µL of a bacterial suspension (0.5 McFarland) and 50 µL of LB Broth in a 96-well round-bottom microplate. The plates were incubated in a microplate reader for 12 h at 37 °C while monitoring the optical density (OD_600_) every 30 min. The absorbance of each well was compared with positive control (50 µL bacteria + 100 µL LB broth) and negative controls (50 µL sewage + 100 µL LB broth).

For enrichment, 10 mL of the sewage combination corresponding to the well showing reduced absorbance was mixed with 2 mL of 0.5 MacFarland bacterial suspension and 2 mL of LB Broth, and incubated overnight at 37 °C. The culture was centrifuged at 4000× *g* for 15 min, and the supernatant was filtered through a 0.2 µm syringe filter. Plaque assay: the filtrate was serially diluted in LB Broth, mixed with 1 mL of 0.5 MacFarland bacterial suspension and 3 mL top agar (semi-solid 0.6% agar), and poured over LB agar plates. Plates were incubated overnight at 37 °C. A single isolated plaque was picked, suspended in 0.5 mL SM buffer (SM buffer: 100 mM NaCl, 8 mM MgSO_4_·7H_2_O, and 50 mM Tris-HCl [1 M, pH 7.5]) centrifuged at 8000× *g* for 5 min to pellet debris, and 100 µL of the supernatant was inoculated into 10 mL LB Broth containing 2 mL of 0.5 MacFarland bacteria suspension and incubated overnight at 37 °C [18]. These steps were repeated 3 times to ensure phage purification. Purified bacteriophage stocks were stored at 4 °C. Plaque assays were performed in triplicate, and plaques were counted on the plate containing 30 to 300 plaques to determine the phage titer (PFU/mL) according to the following formula [20]:$$\mathrm{PFU}/\mathrm{mL}\;=\;\frac{\mathrm{Number}\;\mathrm{of}\;\mathrm{plaques}\;\times \;\mathrm{Dilution}\;\mathrm{factor}}{\mathrm{Volume}\;\mathrm{plated}\;\left(\mathrm{mL}\right)}$$

### 2.3. Bacteriophage Bacteriolytic Activity

Four multiplicity of infection (MOIs) were tested: 10, 1, 0.1, and 0.01, by mixing 100 µL bacteriophage lysate (whose PFU/mL were adjusted for different MOI) with 100 µL 0.5 MacFarland bacterial suspension. The mixtures were prepared in duplicate in a 96-well round-bottom microplate and incubated in a plate reader for 12 h at 37 °C, with the OD at 600 nm recorded every 30 min. Positive control wells contained 100 µL bacteria and 100 µL LB Broth, whereas negative control wells contained 100 µL Phage suspension and 100 µL LB Broth [21].

### 2.4. Host Range

In total, 100 µL of bacteriophage lysate was added to 900 µL of LB Broth and 1 mL of a 0.5 MacFarland bacterial suspension. The mixture was combined with 3 mL LB of top agar (semi-solid 0.6% agar), poured into LB agar plates, and incubated overnight at 37 °C.

### 2.5. One-Step Growth Curve

A total of 15 mL of a 0.5 MacFarland bacterial suspension was prepared, from which 900 µL aliquots were distributed into fifteen 1.5 mL microcentrifuge tubes labeled according to the sampling time points (5, 10, 15 … 70 min). Starting with the 70 min tube, 100 µL of the prepared bacteriophage stock was added sequentially at 5 min intervals, proceeding in reverse order until the 5 min tube. All tubes were then centrifuged at 8000× *g* for 5 min at 4 °C, and the supernatants were carefully collected, kept on ice, serially diluted and plated on LB agar plates. For plating, 1 mL of a 0.5 MacFarland bacterial suspension and 3 mL of LB top agar (semi-solid 0.6% agar) were added to each dilution before pouring onto LB plates, which were incubated overnight at 37 °C. Plaques were counted to determine the phage titers (PFU/mL) at each time point. The burst size was then calculated according to the following formula [22]:$$\mathrm{Burst}\;\mathrm{size}=\frac{\mathrm{total}\;\mathrm{number}\;\mathrm{of}\;\mathrm{phages}\;\mathrm{at}\;\mathrm{the}\;\mathrm{end}\;\mathrm{of}\;1\;\mathrm{cycle}\;}{\mathrm{total}\;\mathrm{number}\;\mathrm{of}\;\mathrm{infected}\;\mathrm{bacteria}\;}$$

### 2.6. Adsorption Rate Assay

Adsorption rate assays were performed following the same procedure as the one-step growth curve, except the incubation time was shortened to 30 min instead of 70. An additional time-zero control was included, in which no bacteria were added- only 900 µL of LB Broth and 100 µL of bacteriophage lysate. For adsorption testing against resistant strains, 100 µL of bacteriophage stock was mixed with 900 µL the bacterial suspension and incubated under the same conditions. At time points 0 and 15 min, samples were centrifuged at 8000× *g* for 5 min at 4 °C, and the supernatants were collected and plated to compare the PFU counts and determine the percentage of adsorbed phages.

### 2.7. Bacteriophage Thermal and pH Stability

Bacteriophage thermal stability was performed at 4, 22, 37, 50, 60, 65, 70, 75, and 80 °C by incubating 250 µL of bacteriophage lysate (diluted in LB Broth) for 2 h. The same procedure was performed to determine the extended stability at 65 °C and 70 °C for 5 and 4 h, respectively.

Three purified and characterized bacteriophages from our lab: EPIMAM01 (*E. coli* ATCC targeting bacteriophage; PQ493298), Miniara127 (*Klebsiella pneumoniae* targeting bacteriophage; PX094878), and AUBFM01 (*Acinetobacter baumannii* targeting bacteriophage; PX094877) were used for comparison. The bacteriophages were diluted to approximately the same initial titer, and the stability test was performed as described above.

pH stability was determined using buffered LB broth adjusted to different pH values. One hundred microliters of bacteriophage lysate were added to 900 µL of buffered LB Broth and incubated for 2 h at 37 °C, followed by a conventional plaque assay to determine the PFU count.

### 2.8. Bacteriophage Genome Sequencing and Analysis

The bacteriophage genomic DNA was extracted using the Norgen Biotek Phage DNA Extraction Kit (Norgen Biotek, Thorold, ON, Canada) according to the manufacturer’s instructions and quantified with a Qubit™ Fluorometer using the Qubit™ dsDNA HS Assay Kit (Thermo Fisher Scientific, Waltham, MA, USA). The DNA was sequenced using the Nextera DNA Library Preparation Kit (Illumina, San Diego, CA, USA) with paired-end reads (2 × 250 bp) on a MiSeq system (Illumina) at the DNA Sequencing Facility of the American University of Beirut. Raw paired-end reads were quality-checked using FastQC, and low-quality bases and adapter sequences were trimmed using Trim Galore. The resulting high-quality reads were assembled de novo with SPAdes, and the assembly quality was assessed using QUAST. The main high-coverage contig was selected for downstream analyses. Genome annotation was performed using Pharokka, and ViPTree (https://www.genome.jp/viptree) was used on 1 october 2025 to construct a viral proteomic tree based on whole-genome sequence similarities [23]. The PhageLeads server (https://phageleads.dk/) was assessed on 1 october 2025 to predict the presence of toxins, virulence, or antimicrobial resistance genes [24]. To calculate the intergenomic distance between the isolated phage and closely related *Salmonella*-targeting bacteriophages, the Virus Intergenomic Distance Calculator (VIRIDIC) (https://rhea.icbm.uni-oldenburg.de/viridic/) was used on 9 November 2025 [25].

### 2.9. Statistical Analysis

For experiments involving multiple groups, data were analyzed using two-way analysis of variance (ANOVA). When a significant interaction or main effect was observed (*p* < 0.05), Tukey’s honest significant difference (HSD) post hoc test was applied to identify specific differences between group means.

## 3. Results

### 3.1. Bacteriophage Isolation and Purification

Twenty-three sewage samples from different sites in Lebanon were tested in eight pooled groups, each representing a combination of three samples, against MDR *Salmonella* Enteritidis strains. The optical density at 600 nm (OD_600_), reflecting bacterial growth, was monitored for 12 h. The sewage combination from Ain EL-Mreiseh, Baabda, and Baalbak showed a significant inhibition of *Salmonella* Enteritidis strain 286 compared with the positive control, suggesting possible lytic activity. To validate this observation, the active sewage combination was enriched overnight with the same host strain to increase phage particle concentration and amplify rare phages. A plaque assay was then performed, revealing large, clear plaques with an average diameter of 3 mm (Figure 1), indicative of bacteriophage presence. The clear plaque morphology is characteristic of lytic bacteriophages, whereas temperate or lysogenic phages typically produce turbid plaques with halos [26]. A single plaque was isolated and enriched overnight with its host, followed by filtration to remove bacterial debris. The resulting lysate was serially diluted and replated, and this procedure was repeated three times to ensure phage purity and genetic homogeneity. The resulting isolate was designated Salmonella phage SA01.

### 3.2. Bacteriolytic Activity

The bacteriolytic activity of Salmonella phage SA01 was evaluated against its host at four MOIs: 0.01, 0.1, 1, and 10, by monitoring bacterial growth over a 12 h period. In all cases, SA01 significantly suppressed bacterial growth during the first six hours, but subsequent regrowth was observed, consistent with the emergence of bacteriophage-resistant *Salmonella* variants (Figure 2). Interestingly, the four tested MOI ratios produced similar inhibition patterns, suggesting that SA01 replicates more rapidly than its bacterial host, thereby minimizing the influence of MOI on lytic activity.

### 3.3. Host Range

The host range of the phage SA01 was assessed using the plaque assay method with undiluted phage lysate. SA01 was able to infect and lyse twelve MDR *Salmonella* Enteritidis strains, whereas seven strains were resistant (Figure 3). Adsorption assay performed on the seven resistant strains revealed that SA01 successfully adsorbed to all seven strains but failed to replicate within them. This broad host range highlights the potential of SA01 for applications in phage therapy, as well as in quality control and biotechnological settings, given the inherent specificity of bacteriophages.

### 3.4. Bacteriophage–Host Interaction Dynamics

SA01 exhibited a high adsorption rate; approximately 97% of the phage particles adsorbed to the host within 3 min, and more than 99% had adsorbed after 5 min. SA01 displayed a short latent period, followed by a rapid rise phase between ~25–45 min, after which the phage titer reached a plateau of approximately 3.2 × 10^9^PFU/mL at ~50–70 min. The calculated burst size was approximately ~32 phage particles per infected cell (Figure 4).

### 3.5. Thermal and pH Stability

The thermal stability of SA01 was evaluated at different temperatures over a two-hour period. SA01 demonstrated remarkable stability at 65 °C, showing only a minimal reduction in viable bacteriophage counts (Figure 5a). Notably, infectious particles remained detectable even at 80 °C, highlighting the strong heat tolerance of this phage. To further confirm this finding, SA01 was incubated for an extended period of 5 h at 65 °C and compared with 3 other purified and characterized bacteriophages: EPIMAM01 (*E coli* ATCC bacteriophage), Miniara127 (*Klebsiella pneumoniae* bacteriophage), and AUBFM01 (*Acinetobacter baumannii* bacteriophage). Prior to incubation, all bacteriophage suspensions were adjusted to approximately the same initial titer (PFU/mL). After 5 h of incubation at 65 °C, only a minimal reduction in SA01 viability was observed, whereas the three reference bacteriophages exhibited a marked loss of infectivity under the same conditions (Figure 6a). To further assess the thermal inactivation and determine the rate of decay of titer decay, SA01 was incubated at 70 °C for 4 h. Under these conditions, the SA01 titer decreased markedly, showing an inactivation slope of ~−0.55 log_10_/h and an average reduction time of ~1.8 h (Figure 6b). These findings highlight the exceptional thermal stability of SA01, which markedly exceeds that reported for most bacteriophages, as viability is typically lost above 60–65 °C. The ability of SA01 to remain infectious even after prolonged exposure to 65 °C suggests a high level of structural robustness and underscores its potential utility in bacteriophage-based food safety strategies and biotechnological applications.

The pH stability of SA01 was tested over a range of pH 4 to 13 by incubating phage suspensions at 37 °C for two hours and determining viable PFU counts. SA01 remained stable across a broad range from pH 5 to 12; however, a marked decrease in phage viability was observed at pH values below 4 (Figure 5b), indicating sensitivity to highly acidic conditions.

### 3.6. Whole-Genome Analysis

Whole-genome sequencing of SA01 yielded a contig of 43,416 bp with an average coverage of ×840. The genome has a GC content of 49.96% and was annotated using Pharokka, which identified 66 open reading frames (ORFs). Among these, 37 ORFs (56.06%) were assigned putative functions, while 29 (43.94%) were annotated as hypothetical proteins (Figure 7a). Functional annotation analysis identified genes associated with DNA replication, structural assembly, and host cell lysis, but no genes related to lysogeny, virulence, or antimicrobial resistance were detected. Genome screening using the PhageLeads server similarly did not identify any toxin, virulence, antimicrobial resistance genes, or predicted temperate lifestyle genes, supporting that SA01 is a lytic bacteriophage and highlighting its potential use as a safe candidate for therapeutic and biocontrol applications. To determine the taxonomic position of bacteriophage SA01, a phylogenetic tree was generated using viptree with the 18 most closely related bacteriophages: *Salmonella* phage blauehaus (NC_073173), *Salmonella* phage NBSal007 (NC_073183), *Salmonella* phage, Shelanagig (NC_073195), *Salmonella* phage NBSa1006 (NC_073196), *Salmonella* virus VSe101 (NC_073176), *Salmonella* phage vB_StyS-sam (NC_073177), *Salmonella* phage vB_SenS_TUMS_E4 (NC_073179), *Salmonella* phage vB_SenS_TUMS_E19 (NC_073178), *Salmonella* phage wast (NC_073174), *Salmonella* phage skrot (NC_073172), *Salmonella* phage vB_SenS-Ent2 (NC_023608), *Salmonella* phage vB_SenS-Ent3 (NC_024204), *Salmonella* phage vB_SenS-Ent1 (NC_019539), *Salmonella* phage vB_SenS_ER23 (NC_073197), *Salmonella* phage vB_SenS_ER21 (NC_073198), *Salmonella* phage dunkel (NC_073201), *Salmonella* phage S102 (NC_073199), *Salmonella* phage S100 (NC_073200) (Figure 7b). This analysis indicates that Salmonella phage SA01 belongs to the genus *Jerseyvirus*, subfamily *Guernseyvirinae*, Family *Sarkviridae*, and Class *Caudoviricetes*. Genomic comparisons (tBLASTx) were made between Salmonella phage SA01, and the closely related phage; *Salmonella* virus VSe101 (NC_073176) performed using viptree software revealed multiple locally collinear blocks with >95% nucleotide identity, indicating a high level of conservation in gene content, along with several deletions, insertions, translocations, and small rearrangement events at specific loci (Figure 7c). To further assess taxonomic relatedness, the genome of Salmonella phage SA01 (43,416 bp) was compared with representative *Jerseyvirus* genomes using the VIRIDIC tool [27]. Pairwise nucleotide similarities ranged from 90.1% to 93.9% with *Salmonella* phages vB_SenS-Ent1, and NBSal007, corresponding to intergenomic distances of 0.06–0.10. These values fall below the 95% species demarcation threshold but above the 70% genus threshold, indicating that SA01 represents a novel species within the genus *Jerseyvirus*, family *Sarkviridae*. The SA01 genome has been submitted to the GenBank database under accession number PX246135 (Salmonella Phage SA01 Genome and annotation are included in the Appendix A).

## 4. Discussion

It is beyond question that the antimicrobial resistance (AMR) crisis has become a major global threat. As the number of multidrug-resistant (MDR) strains and MDR-associated mortality continue to increase each year, experts warn that we are entering a post-antibiotic era [28]. In this context, and given that nearly 70% of foodborne infections are caused by food-transmitted pathogens, effective strategies to prevent and control these microorganisms are critical to ensuring food safety and public health [29]. Among these pathogens, Salmonella ranks among the leading causes of foodborne illnesses. According to the European Food Safety Authority and the European Centre for Disease Prevention and Control, Salmonella is the second most frequent cause of foodborne infection in the European Union [30].

Consistent with this global concern, an ongoing study at the Department of Experimental Pathology, Immunology & Microbiology, Center for Infectious Diseases Research, and WHO Collaborating Centre for Reference & Research on Bacterial Pathogens at the American University of Beirut has reported a significant increase in antibiotic resistance among Salmonella Enteritidis isolates in Lebanon over the past two decades. Considering that poultry meat has reached an annual global consumption of about 140 million tons, making it the most consumed meat worldwide, there is an urgent need to develop novel strategies to control these pathogens [31].

In light of these challenges, bacteriophages have emerged as promising candidates for use as antibiotic alternatives in both therapeutic and food safety applications. Because they are highly specific, self-replicating, and target MDR bacteria, phages exert minimal effects on human cells or the commensal microbiota while efficiently lysing their specific bacterial hosts. In contrast to antibiotics, which act broadly and often disrupt beneficial microbiota, phages recognize their hosts through precise receptor interactions before DNA injection and infection, thereby sparing non-target bacterial populations.

Salmonella phage SA01 exhibited strong lytic activity, lysing twelve out of nineteen tested MDR *Salmonella* Enteritidis clinical isolates obtained from human stool samples. Its extraordinary thermal stability and rapid replication make it a promising biological agent for therapy, food safety, and biotechnology. Sewage served as an excellent source for phage isolation since it is rich in both bacteria and their corresponding parasites. Samples were screened in groups of three to increase the likelihood of isolating a *Salmonella*-specific phage, leading to the isolation of three phages from eight screened combinations. SA01 was isolated from a mixture collected from Ain El-Mreisseh, Baabda, and Baalbek. It produced large, clear plaques, suggesting its lytic life cycle, which was later confirmed by genomic analysis. SA01 also showed a fast replication cycle and high diffusion rate in semi-solid agar (0.6% LB agar). Because genetic homogeneity is essential for downstream experimental reliability, plaques were picked and re-enriched three times. A plaque results from infection of a single bacterium by a single infectious particle; as the phage replicates and lyses the host cells, it forms a clear zone through sequential infection cycles. Therefore, all phages within one plaque are genetically identical, aside from possible spontaneous mutations [32].

SA01 successfully adsorbed to all tested MDR *Salmonella enterica* strains and lysed more than half of them. The resistant strains likely possess intrinsic defense mechanisms against SA01, such as CRISPR-Cas systems or restriction–modification enzymes. The SA01 receptor appears to be widely conserved across diverse *Salmonella* strains, as indicated by its rapid adsorption rate and broad host range despite genetic diversity among the tested isolates (Figure 3). This broad host range supports its potential as a biocontrol and therapeutic phage.

The bacteriolytic activity assay showed that SA01 strongly inhibited bacterial growth during the first six hours, after which bacterial regrowth occurred, consistent with the emergence of resistant variants, a common limitation of single-phage treatments. This underscores the importance of using phage cocktails for effective biocontrol. All tested MOIs displayed similar inhibition profiles, suggesting that SA01 replicates faster than its host, allowing efficient infection even at low phage-to-bacterium ratios (MOI = 0.01). These findings highlight its strong lytic potential and suggest that low concentrations may be sufficient for effective control.

SA01 exhibited a rapid adsorption rate, reflecting high affinity for its receptor, along with a short latent period of 20 min and a burst size of approximately 32 PFU per infected cell. This combination allows the phage to effectively compete with bacterial replication and accounts for its large plaque size and strong lytic activity, even at low MOI. Although the burst size of 32 particles per infected cell is lower than that of previously reported Salmonella phages such as PSDA-2 (S. Typhimurium, 120 PFU/cell) [33] and LPSE1 (S. enterica, 94 PFU/cell) [34], SA01’s rapid adsorption rate and short latent period compensate for this difference, enabling efficient bacterial suppression and supporting its potential as a biocontrol agent against MDR Salmonella.

While most bacteriophages lose viability within minutes at 60–65 °C [35,36,37], SA01 maintained infectivity at 65 °C for up to five hours with only minimal reduction in viable particles. Moreover, active phages were still detectable after exposure to 80 °C for two hours. This remarkable stability suggests a uniquely robust capsid structure capable of preserving infectivity under extreme conditions where most mesophilic bacteria and phages are inactivated. Such thermostability enables SA01 to withstand pasteurization and other mild heat treatments used in food processing, expanding its applicability in the food industry. Its additional stability over a wide pH range (3.7–11.3), encompassing the pH values of most foods, further supports its use in diverse environments, including raw and ready-to-eat foods, without the need for chemical preservatives.

Given that poultry production and animal feed pellets are often processed at 60–70 °C, SA01 could be incorporated to achieve synergistic effects between heat and phage lysis. In the near future, thermostable phage cocktails may provide sustainable alternatives to antibiotics in agriculture and food production, helping curb AMR emergence. In addition, thermostable phages such as SA01 could be integrated into sanitation protocols targeting food-contact surfaces and equipment, which often serve as reservoirs for *Salmonella* transmission.

Beyond food safety, the thermostable capsid proteins of SA01 could have biotechnological applications. Phages have long been used as cloning vectors, and thermostable variants could expand the range of molecular and synthetic biology tools. Moreover, thermostable phages could improve the stability and shelf life of biosensors or rapid pathogen detection assays, as they are less susceptible to inactivation by environmental stressors such as temperature fluctuations, light exposure, or long-term storage.

Genome annotation of SA01 revealed genes encoding key structural and functional proteins, including the major capsid protein, tail sheath, tail fiber, DNA polymerase, endolysin, holin, and spanins. However, capsid thermostability does not necessarily imply thermostability of all encoded enzymes; while the capsid protects the genome under harsh conditions, phage enzymes are typically adapted to the physiological temperature of the bacterial host. Importantly, in silico screening by Pharokka and PhageLeads revealed no lysogeny, virulence, or AMR-associated genes, confirming that SA01 follows a strictly lytic life cycle and is safe for use, unlike temperate phages, which may mediate horizontal gene transfer and spread resistance genes.

Although bacteria can develop resistance to phages, unlike antibiotics, phages possess the capacity to coevolve alongside their hosts, adapting to overcome bacterial defenses. Nonetheless, if phages were ever to replace antibiotics and be used indiscriminately, this could alter microbial community dynamics and exert unforeseen ecological consequences. Therefore, rational use, continuous surveillance, and adherence to regulatory frameworks will be essential as phage-based strategies move toward clinical and industrial implementation.

## 5. Conclusions

In this study, we have isolated a novel MDR *Salmonella* Enteritidis-targeting bacteriophage, designated Salmonella phage SA01, from Lebanese sewage. SA01 exhibited desirable biological properties, including extraordinary thermal stability, broad host range, and strong lytic activity. Whole-genome analysis confirmed the absence of lysogeny, virulence, and antimicrobial resistance genes, ensuring its safety for therapeutic and food safety applications. Collectively, these features highlight SA01 as a promising candidate for bacteriophage therapy, food safety interventions, and potential biotechnological applications; however, further research is essential to validate the efficacy and safety of SA01 under practical conditions.

## Figures and Tables

**Figure 1 viruses-17-01518-f001:**
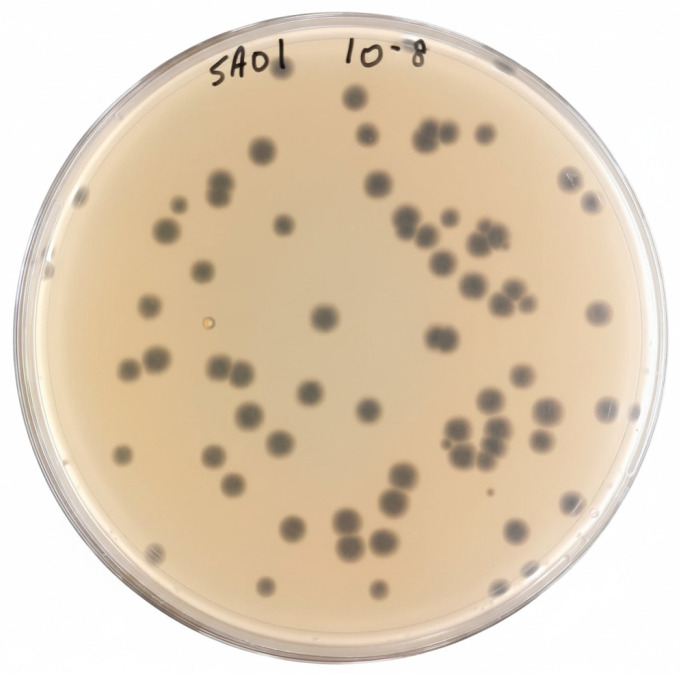
Morphological characteristics of Salmonella phage SA01. Plaque morphology observed on a double-layer agar plate. on the plate: SA01 phage, dilution: 10^−8^.

**Figure 2 viruses-17-01518-f002:**
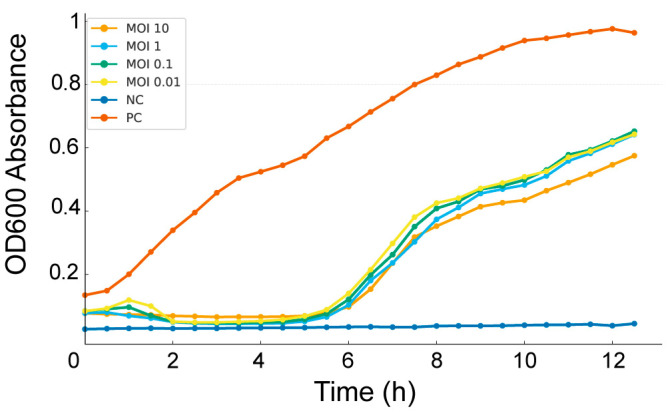
Bacteriolytic activity of Salmonella phage SA01 against its host at different multiplicities of infection (MOIs). Bacterial growth was monitored by measuring OD_600_ absorbance over 12 h. In the absence of bacteriophage (positive control PC, orange), bacteria grew steadily to high densities. In contrast, bacteriophage-infected cultures showed dose-independent inhibition of bacterial growth. Bacterial regrowth occurred after an initial inhibition phase, indicating partial control by the bacteriophage. No increase in OD_600_ absorbance in the absence of bacteria in the negative control NC.

**Figure 3 viruses-17-01518-f003:**
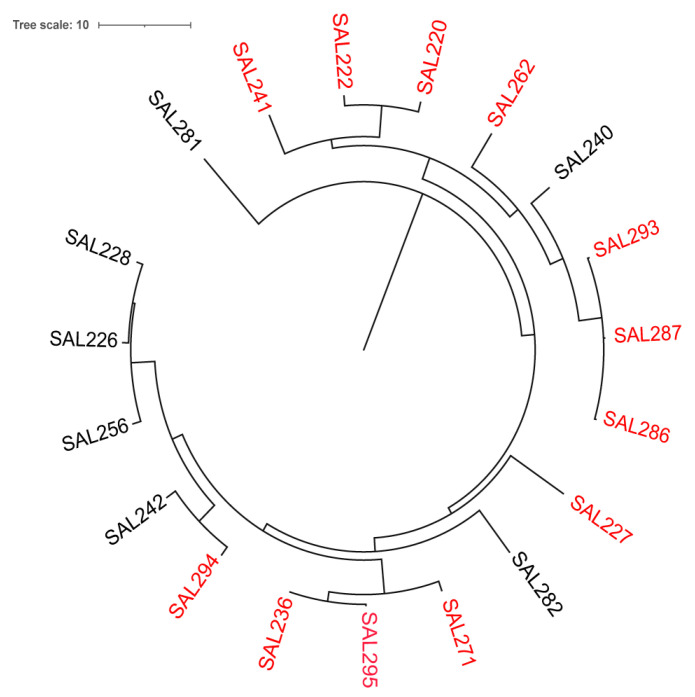
Host range analysis of Salmonella phage SA01 across diverse clinical and environmental isolates. A circular phylogenetic tree was generated based on *Salmonella* strains, with susceptible isolates (red labels) indicating successful bacteriophage infection and resistant isolates showing no detectable lytic activity. The distribution of susceptible strains across multiple clades highlights the broad host range of SA01 within the *Salmonella* genus.

**Figure 4 viruses-17-01518-f004:**
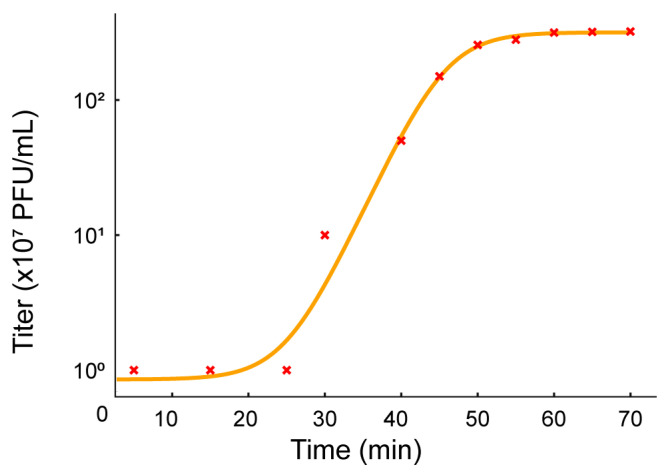
Biological characterization of Salmonella phage SA01. One-step growth curve of Salmonella phage SA01 fitted with a 4-parameter logistic model. Titers (×10^7^ PFU/mL) were measured at multiple time points during a one-step growth experiment and plotted on a logarithmic scale. The fitted curve reveals an early latent period followed by a sharp burst between ~25–45 min, reaching a plateau at ~50–70 min.

**Figure 5 viruses-17-01518-f005:**
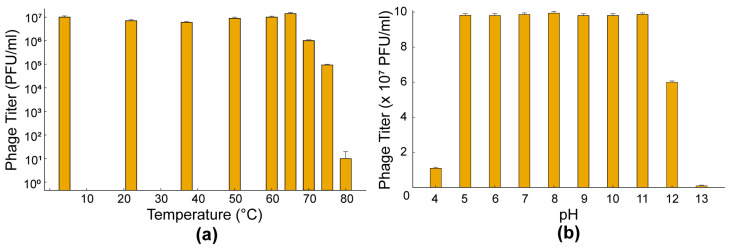
Thermal and pH Stability Profiles of the Isolated Bacteriophage. (**a**) Thermal stability of the bacteriophage was assessed over a temperature range from 4 °C to 80 °C and presented as bacteriophage titer (PFU/mL Log scale). The bacteriophage maintained full stability up to ~65 °C, after which a sharp decline in viability was observed; (**b**) pH stability of the bacteriophage across a range of pH 4–12. The bacteriophage showed high stability in the neutral to slightly alkaline range (pH 6–9), with decreased viability under strongly acidic or alkaline conditions.

**Figure 6 viruses-17-01518-f006:**
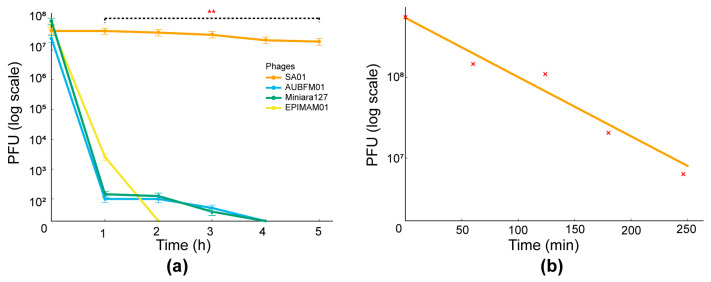
Comparative thermal stability and inactivation kinetics of Salmonella phage SA01. (**a**) Thermal stability of four bacteriophages at 65 °C. Bacteriophage titers (PFU/mL, log scale) were monitored over a 5 h incubation period. Among the tested bacteriophages, only SA01 (orange) retained infectivity throughout the assay, demonstrating remarkable thermostability. In contrast, AUBFM01 (blue), Miniara127 (green), and EPIMAM01 (yellow) showed rapid loss of activity, with complete inactivation within 2–3 h. Statistical analysis was performed using two-way ANOVA (factors: bacteriophage and time), followed by Tukey’s post hoc test. Results confirmed a highly significant effect of both bacteriophage identity and incubation time (** *p* < 0.001), with SA01 showing statistically greater stability than all other bacteriophages across all time points. (**b**) Thermal inactivation kinetics of Salmonella phage SA01 at 70 °C. Decay of phage titers (PFU/mL, log scale) was determined at different time intervals over 4 h at 70 °C. A linear regression model fitted to log-transformed data revealed a steady decline in phage infectivity over time, with an inactivation slope of ~−0.55 log_10_/h. SA01 followed first-order thermal inactivation kinetics, with an average reduction time of ~1.8 h, indicating an exponential decay in titer that was independent of the initial concentration.

**Figure 7 viruses-17-01518-f007:**
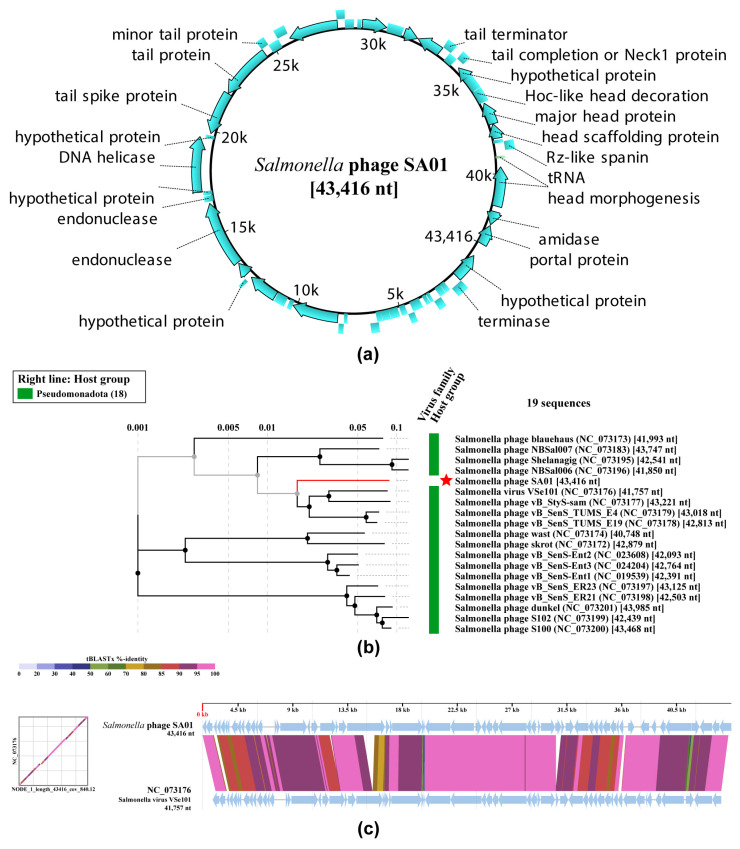
Genomic features and phylogenetic analysis of Salmonella phage SA01. (**a**) Circular genome map of SA01 (43,416 bp) showing predicted open reading frames (ORFs) and their annotated functions; (**b**) Phylogenetic tree based on whole-genome nucleotide similarity, showing SA01 (red star) clustering with closely related *Salmonella* bacteriophages. The tree includes 18 reference sequences retrieved from GenBank, and branch lengths indicate genetic distance; (**c**) Whole-genome alignment of SA01 against its closest relatives using ViPTree. Conserved locally collinear blocks are shown as colored segments, highlighting overall genome synteny with evidence of rearrangements and unique regions distinguishing SA01.

## Data Availability

The original contributions presented in this study are included in the article and Appendix A; further inquiries can be directed to the corresponding author.

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
