# Peer review of "Isolation and Characterization of a Novel Thermostable Bacteriophage Targeting Multi-Drug-Resistant Salmonella Enteritidis"

_viruses, 2025, doi:10.3390/v17111518_

Round 1

Reviewer 1 Report

Comments and Suggestions for Authors

Dear Authors,

The manuscript, “Isolation and Characterization of a Novel Thermostable Bacteriophage targeting Multi-Drug-Resistant Salmonella Enteritidis,” submitted for review, presents the phage characteristics clearly and concisely. The research methods used were fully justified, and the results were presented in accordance with generally accepted requirements.

The most important concern is that it is not possible to find a publicly accessible direct link to the GenBank record with accession number PX246135 for Salmonella phage SA01. Please add the correct link in the manuscript.

Below, I  present my minor comments.

I am not sure why double Salmonella strain names are given in Materials and Methods (in table A1 everything is repeated)

verse 174: Please, explain what is AGI?

Figure 1: The plate is labeled as AGI295, but Salmonella Strain 286 is described as the host of phage SA01. So, which strain is the host of the phage?

Figure 2: The figure description is insufficient. The authors need to explain what “NC” and “PC” refer to.

Authors should use the intergenomic distance calculator (VIRIDIC).

Table A1, A2.1, A3 – Please provide a legend for all tables and define all abbreviations used

References 15: should be the original FDA document here

The authors should use the correct notation when reporting pH, and the Latin name Salmonella should be written in italics.

The manuscript would benefit from careful language revision, particularly regarding the consistent use of verb tenses.

I recommend the manuscript for publication after minor revision

Best regards,

Author Response

Comment 1: The most important concern is that it is not possible to find a publicly accessible direct link to the GenBank record with accession number PX246135 for Salmonella phage SA01. Please add the correct link in the manuscript.

Response 1: This accession number was provided by GenBank upon submission. We are also surprised why it's not accessible yet. In the meantime,  we will include the phage genome sequence and annotation as supplementary material (in FASTA and genbank format) while the GenBank record is being made publicly accessible.

Comment 2: I am not sure why double Salmonella strain names are given in Materials and Methods (in table A1 everything is repeated)

Response 2: The names will be removed from the materials and methods section. 

Comment 3: verse 174: Please, explain what is AGI?

Respond 3: Sorry for the confusion, AGI stands for the exact locations of sewage collection, Ain EL-Mreiseh, Gaboon in baabda and Iaat in Baalbek. We will remove this abbreviation and mention only the full name.

Comment 4: Figure 1: The plate is labeled as AGI295, but Salmonella Strain 286 is described as the host of phage SA01. So, which strain is the host of the phage?

Respond 4: Yes, SAL295 is susceptible to Salmonella phage SA01, but this image was mistakenly uploaded instead of its used host, SAL286. We will provide another clear image of plaques.

Comment 5: Figure 2: The figure description is insufficient. The authors need to explain what “NC” and “PC” refer to.

Respond 5: Sure, we will emphasize more in the legend, as the PC is the positive control where bacteria without bacteriophages are incubated, and NC is the negative control.

Comment 6: Authors should use the intergenomic distance calculator (VIRIDIC).

Response 6: We thank the reviewer for this valuable suggestion. Intergenomic distances were calculated using VIRIDIC (Rhea et al., BMC Genomics, 2021), the ICTV-recommended tool for prokaryotic virus classification. The analysis included our phage (43,416 bp) and representative Jerseyvirus members from the family Sarkviridae. The results showed 90–94 % nucleotide identity (intergenomic distances = 0.06–0.10) between our phage and its closest relatives, including Salmonella phages vB_SenS-Ent1 (HE775250.1). According to ICTV demarcation thresholds (> 95 % identity for species, > 70 % for genus), these values confirm that our isolate represents a new species within the genus Jerseyvirus.

Comment 7: Table A1, A2.1, A3 – Please provide a legend for all tables and define all abbreviations used.

Respond 7: Sorry for the confusion, we will add Legends for each table and define the used abbreviations.

Comment 8: References 15: should be the original FDA document here

Respond 8: We have referenced an FDA Agency Response Letter GRAS Notice No. GRN 000603.

Comment 9: The authors should use the correct notation when reporting pH, and the Latin name Salmonella should be written in italics.

Response 9: The comment will be applied. 

Thank you for your valuable review.

Reviewer 2 Report

Comments and Suggestions for Authors

The study focuses on characteristic of thermo-stable phage infecting S. Enteritidis. The overall experiment design is done in accordance with well established protocols and research papers. Hovewer, I do have remarks regarding quality of the work's presentation.

First of all, in materials and methods section some methods are described together (i.e. one step growth and kinetics of adsorption) making it difficult to understand what has actually been done at which point in time. Please revise.

Also, there are no citations in matherial and methods section. Why? The methods are well established and protocols are easily available. Please reference the works based on witch you planned your own experiments.

As the stability tests are not done as one sample being dragged through different temperatures over time (or one sample heated up or cooled down and measured) it should not be presented as linear graph, bar chart would be more appropriate.

Author Response

comment 1:  in materials and methods section some methods are described together (i.e. one step growth and kinetics of adsorption) making it difficult to understand what has actually been done at which point in time. Please revise.

response 1: We will separately describe the following methods, and check if there are others, for better representation and more simplicity.

comment 2: there are no citations in material and methods section. Why? The methods are well established and protocols are easily available. Please reference the works based on which you planned your own experiments.

response 2: Sorry for the confusion, sure we will include references for each method.

comment 3: As the stability tests are not done as one sample being dragged through different temperatures over time (or one sample heated up or cooled down and measured) it should not be presented as linear graph, bar chart would be more appropriate.

response 3: Sure, the stability data will be represented as bar chart rather than linear graph. 

Thank you for your valuable review.

Round 2

Reviewer 2 Report

Comments and Suggestions for Authors

The authors have made the changes I have requested during the first round. However, I still have some minor suggestions regarding the presentation of the results.

  1. In graphs like in fig 2 or 4 make sure that both "0" are at the beggining of the axis, in the same point (where the both axis begin).
  2. Figure 5 - you have changed it into a bar chart, but I do not see any SD bars, nor statistical analysis.

Author Response

Comment 1: In graphs like in fig 2 or 4 make sure that both "0" are at the beggining of the axis, in the same point (where the both axis begin)

Response 1: We'll ensure the graphs are modified accordingly.

Comment 2: Figure 5 - you have changed it into a bar chart, but I do not see any SD bars, nor statistical analysis. 

Respond 2: No statistical analysis was performed for the temperature and pH stability assays, as these experiments were intended to assess qualitative stability trends rather than quantitative differences. The decrease in phage titer under certain conditions was evident and consistent across replicates, confirming the expected loss of stability without requiring formal statistical testing.

Thank you very much for your valuable review.